# Unilateral Post-Chemotherapy Robot-Assisted Retroperitoneal Lymph Node Dissection for Stage II Non-Seminomatous Germ Cell Tumors: Sexual and Reproductive Outcomes

**DOI:** 10.3390/cancers16122231

**Published:** 2024-06-15

**Authors:** Antonio Tufano, Simone Cilio, Gianluca Spena, Alessandro Izzo, Luigi Castaldo, Giovanni Grimaldi, Raffaele Muscariello, Dario Franzese, Giuseppe Quarto, Riccardo Autorino, Francesco Passaro, Sisto Perdonà

**Affiliations:** 1Uro-Gynecological Department, Istituto Nazionale Tumori di Napoli, IRCCS “G. Pascale”, Via M. Semmola, 80131 Naples, Italya.izzo@istitutotumori.na.it (A.I.); dario.franzese@istitutotumori.na.it (D.F.); s.perdona@libero.it (S.P.); 2Department of Maternal-Infant and Urological Sciences, Policlinico Umberto I Hospital, “Sapienza” Rome University, 00161 Rome, Italy; 3Department of Neurosciences, Science of Reproduction and Odontostomatology, University of Naples Federico II, 80131 Naples, Italy; 4Department of Urology, Rush University Medical Center, Chicago, IL 60612, USA

**Keywords:** testis cancer, testicular cancer, retroperitoneal lymph node dissection, robotic retroperitoneal lymph node dissection, RPLND, non-seminoma, sexual outcomes

## Abstract

**Simple Summary:**

Post-chemotherapy retroperitoneal lymph node dissection (PC-RPLND) is an integral component of testicular cancer treatment, as approximately one-third of patients will have a residual mass after undergoing chemotherapy. This study aimed to assess sexual and reproductive outcomes in patients undergoing post-chemotherapy robot-assisted retroperitoneal unilateral lymph node dissection (PC-rRPLND) for non-seminomatous germ cell tumors (NSGCTs) at a high-volume cancer center. Preoperative and postoperative (at 12 months) ejaculatory function as well as erectile function, based on the International Index of Erectile Function-5 (IIEF-5) and Erection Hardness Score (EHS), were assessed. Overall, 22 patients were included. This study highlights retrograde ejaculation as a significant complication of PC-rRPLND, along with a non-negligible occurrence of erectile dysfunction among patients.

**Abstract:**

We aimed to report sexual and reproductive outcomes following post-chemotherapy robot-assisted retroperitoneal unilateral lymph node dissection (PC-rRPLND) for non-seminomatous germ cell tumors (NSGCTs) at a high-volume cancer center. We collected records regarding sexual and reproductive outcomes of patients undergoing unilateral PC-rRPLND for stage II NSGCTs from January 2018 to November 2021. Preoperative and postoperative (at 12 months) ejaculatory function as well as erectile function, based on the International Index of Erectile Function-5 (IIEF-5) and Erection Hardness Score (EHS), were assessed. Only patients with a pre-operative IIEF-5 of ≥22 and EHS of ≥3 were included in this analysis. Overall, 22 patients undergoing unilateral PC-rRPLND met the inclusion criteria. Of these, seven (31.8%) patients presented an andrological disorder of any type after PC-rRPLND. Specifically, retrograde ejaculation was present in three (13.6%) patients and hypospermia was present in one (4.5%) patient. Moreover, three (13.6%) patients yielded erectile dysfunction (IIEF-5 < 22 and/or EHS < 3). Lastly, two (9.1%) succeeded in naturally conceiving a child after PC-rRPLND. Retrograde ejaculation is confirmed to be one of the most common complications of PC-rRPLND. Moreover, a non-negligible number of patients experience erectile dysfunction.

## 1. Introduction

Testicular cancer patients with residual masses after platinum-based chemotherapy, as well as those who present with a clinical stage I and II NSGCT, are suitable candidates for retroperitoneal lymph node dissection (RPLND) [1]. The open approach to this procedure has been considered the standard of care for a long period, but is burdened by significant surgical morbidity. While offering a minimally invasive option, the laparoscopic approach is plagued by technical complexity [2,3]. The advent of the robotic RPLND (rRPLND) improved outcomes thanks to its shortened learning curve with improved visualization, instrument dexterity, and better operator ergonomics [4,5,6].

Despite optimal oncological outcomes, the division of the sympathetic plexus during standard rRPLND often translates into sexual dysfunctions, with ejaculatory dysfunctions such as dry or retrograde ejaculation (REj) being the most common [7].

A modified unilateral template for rRPLND allows for avoiding nerve damage and ultimately achieving antegrade ejaculation in 85% of patients [8]. Recently, the unilateral approach was adopted by Mistretta et al., providing a complete analysis of retrospective data regarding sexual outcomes in a series of 32 patients who underwent primary or post-chemotherapy unilateral rRPLND. Overall, more than 30% of patients presented an andrological disorder. In particular, the authors reported hypospermia in 12.5% of patients and rates of 9.4% of both REj and erectile dysfunction (ED) [9].

Since there is still a lack of data and reliable predictors of sexual impairment in post-chemotherapy patients affected by retroperitoneal disease, we sought to assess sexual and reproductive outcomes following unilateral post-chemotherapy rRPLND (PC-rRPLND) for stage II NSGCTs at a single high-volume cancer.

## 2. Materials and Methods

### 2.1. Study Population

We retrospectively analyzed the medical records of testicular cancer patients who had been referred and treated at our tertiary care referral cancer center between January 2018 and November 2021. Socio-demographic and clinical characteristics were recorded for each patient, including body mass index (BMI) and American Society of Anesthesiologists (ASA) score [10]. Before primary orchiectomy, all patients were offered sperm cryo-conservation.

Only patients with histologic diagnosis of a stage II NSGCT were considered for the analysis. Following orchiectomy, certain patients underwent a chemotherapy regimen comprising three cycles of bleomycin-etoposide cisplatin (BEP), administered based on the histological characteristics of the primary tumor and instrumental and biochemical findings. Subsequent to the completion of the chemotherapy regimen, serum tumor marker assessment and thoracoabdominal CT scans were conducted to ascertain the presence of post-chemotherapy residual masses.

Unilateral PC-rRPLND was offered in the case of ipsilateral residual masses of <5 cm in diameter and either normalized or plateauing serum tumor markers in patients. After excluding literacy problems, the International Index of Erectile Function-5 (IIEF-5) for all patient in the preoperative period and after 12 months of follow-up was compiled. The IIEF-5 is a 5-item questionnaire scoring from 5 to 25, with a score <22 indicating compromised erectile function. Simultaneously, the erectile function of each patient was assessed by the Erection Hardness Score (EHS), a single-item, patient-reported outcome by self-scoring erection hardness from 1 to 4, with a score inferior to 3 indicating compromised erectile function. Thus, in order to better elucidate the effects of the procedure on sexual function, only patients with pre-operative IIEF-5 ≥ 22 and EHS scores ≥ 3 were included for further analysis. Moreover, data from patients with secondary RPLND, salvage RPLND (rising serum tumor markers during or after chemotherapy), and RPLND for late relapse (disease recurrence of >2 years after last treatment) were excluded.

The present study was approved by the Ethical Committee of IRCCS Pascale-A.O.R.N. Santobono-Pausilipon (protocol number 56/22) and all patients signed an informed consent form.

### 2.2. Surgical Technique

All surgical procedures were conducted by a singular highly experienced surgeon (SP) specialized in robotic surgery. The transperitoneal approach was uniformly employed across all cases. Surgical interventions utilized either the da Vinci SI or Xi surgical systems (Intuitive Surgical Inc., Sunnyvale, CA, USA).

Briefly, when utilizing the da Vinci SI robotic system, patients were positioned in the flank posture, with a 12 mm camera port and two 8 mm robotic ports configured linearly in the pararectal region. Additionally, two auxiliary ports of 5 mm and 12 mm were employed. Conversely, with the da Vinci Xi robotic system, patients were placed supine on the operating table, with the camera port positioned 1 cm below the umbilicus. Three additional ports were then positioned along the same line at the level of the pararectal line (either right or left depending on tumor location) and of the mid-clavicular right and left lines.

The surgical approach followed the modified template as outlined by Heidenreich et al. for robotic retroperitoneal lymph node dissection (rRPLND) [8]. This approach involved dissection of the left-sided template encompassing preaortic nodes up to the level of the inferior mesenteric artery, para-aortic nodes, and retro-aortic nodes, with the ureteral crossing of the iliac artery demarcating the caudal and lateral boundaries of the dissection. For the right-sided template, resection included paracaval, precaval, retrocaval, and interaortocaval nodes along with the area lateral to the common iliac vessels, with the ureteral crossing serving as the caudal boundary and the ureter delineating the lateral boundary of dissection.

### 2.3. Statistical Analysis

Descriptive statistics were provided for demographics, clinical characteristics, and intraoperative and postoperative parameters. Medians and interquartile ranges or frequencies and proportions were reported for continuous or categorical variables, respectively. Statistical analyses were performed using the RStudio graphical interface v.0.98 for R software environment v.3.0.2 (http://www.r-project.org, accessed on 16 April 2024).

## 3. Results

Overall, 33 patients underwent unilateral PC-rRPLND for stage II NSGCTs between 2018 and 2021. Of those, 22 (66.7%) patients met the inclusion criteria and were included in the analysis (Table 1). Of these, 10 (45.4%) and 12 (54.6%) men underwent right and left primary orchiectomy, respectively. The histological valuation from primary orchiectomy depicted lymph vascular invasion in 15 (68.2%) of patients, with a proliferation rate >70% in 11 (50.0%), 12 (54.5%), and 8 (36.4%) of them presenting >50% embryonal carcinoma and teratoma, respectively. A single event of an intraoperative complication (vena cava injury) was recorded, without needing perioperative blood transfusion. Two episodes of ileus, an early postoperative complication, occurred. None of the patients experienced late post-operative complications.

Table 2 depicts the andrological outcomes of unilateral PC-rRPLND after a median (IQR) follow-up period of 36 (20–42) months. Overall, seven (31.8%) patients reported a new-onset andrological dysfunction (any) after PC-rRPLND. Three (13.6%) men complained of newly developed erectile dysfunction (ED), as identified by IIEF-5 and/or EHS. REj was reported by three (13.6%) patients and hypospermia was present in one (4.5%) patient. Lastly, two (9.1%) patients successfully conceived a child without relying on assisted reproductive techniques (ART) (Table 2).

## 4. Discussion

Our analysis shows that almost one out of three patients undergoing unilateral PC-rRPLND for a stage II NSGCT develop a sexual dysfunction. The most common complaints were ED and REj, followed by hypospermia. Interestingly, two patients reported to have naturally conceived a child during the follow-up, without the need for ART.

NSGCTs have survival rates ranging from 96% to 67% according to histological features and staging. In this setting, BEP-based chemotherapy has been shown to be highly effective in the management of metastatic disease, which is usually located in the retroperitoneum [11]. However, residual retroperitoneal masses following chemotherapy can persist or recur in up to 30%, and here is where RPLND plays an important role in patients’ survival [12,13]. Indeed, previous data report five-year overall survival rates of 89% in patients who underwent PC-rRPLND for a stage II NSGCT [14]. Nonetheless, the complex nature of this surgery is not exempt of intra- and post-operative complications.

In this regard, Gerdtsson et al. published results from a cohort of 318 patients who underwent PC-rRPLND (27% bilateral and 73% unilateral approach, respectively) included in the Swedish and Norwegian Testicular Cancer Group (SWENOTECA)’s prospectively built database. The authors reported that ureteral injuries were the most common intra-operative complications, followed by aorta and inferior vena cava injuries, whilst lymphatic leakage and wound infection were more common during the first postoperative month. The only long-term complication was identified as REj [15]. Similarly, in a previous study, we observed that the unilateral PC-rRPLND approach for NSGCTs was a feasible surgical procedure characterized by low perioperative morbidity and safe oncological outcomes while recording acceptable rates of anterograde ejaculation in 75.8% of patients [16]. Owing to the young age at diagnosis and the overall good prognosis of NSGCTs and considering the impact of erectile and orgasmic dysfunction on the men’s overall quality of life, long-term treatment-related outcomes on sexual function are an important health concern [17,18]. In this regard, the current literature shows rates of REj between 5.5% and 10.5% in patients who underwent PC-rRPLND [19,20]. In a recent systematic review and meta-analysis, Ge et al. report no differences in the post-operative ejaculatory disorders between men who underwent robotic or non-robotic RPLND. However, the authors outlined that only a few studies dealt with ejaculation function and the heterogeneity among study designs was high (I^2^ = 61.1%; *p* < 0.1), thus decreasing the reliability of the results [21]. In this open debate on ejaculatory dysfunction among patients undergoing unilateral PC-rRPLND, our study depicts rates of 13.6% of REj and 4.5% of hypospermia.

In previous studies, ED was identified as a transient occurrence affecting nearly a quarter of patients [22,23]. However, in another study involving complete bilateral non-nerve-sparing PC-RPLND, no discernible disparities were noted in erectile function before and after surgery [24]. Nevertheless, significant impairments in orgasmic function, intercourse, and overall sexual satisfaction were observed following RPLND. In our cohort, more than one in three testicular cancer survivors can experience ED, with higher prevalence rates among those who require more than one treatment modality (radiation, surgery, and chemotherapy) [25]. However, only a few published data explore the prevalence of new-onset ED in men who underwent RPLND. In a retrospective study by Dimitropoulos et al. involving 53 patients who underwent PC-rRPLND, no differences were found in terms of new-onset ED before and after the surgery [24]. In this scenario, our analysis depicted that a non-neglectable percentage of men were diagnosed with ED, which was equal to that of REj. Consequently, as a major point of strength of our study, our data on ED prevalence were recorded among patients with pre-unilateral PC-rRPLND normal erectile function, thus highlighting the specific impact of this procedure and excluding the burden derived from the diagnosis of testicular cancer itself and its primary treatment.

Nonetheless, due to the high incidence during reproductive age, men with testicular cancer have to deal with fertility issues [26]. Indeed, on the one hand, previous findings showed that even before their diagnosis, patients with testicular cancer can display an alteration in terms of sperm count and motility, which can be deteriorated by orchidectomy and chemotherapy [27,28]. On the other hand, the multimodal therapeutic approach reserved to these patients can worsen these parameters, with the lowest fertility rates reported in patients treated with combined therapy, thus including men who had undergone PC-rRPLND [22,29]. In this regard, our study depicted that two patients successfully achieved a pregnancy, thus indicating that the adopted unilateral approach could preserve fertility in almost 10% of men with a stage II NSGCT.

Open retroperitoneal lymph node dissection (o-RPLND) is the accepted standard surgical approach to treat retroperitoneal nodal disease in testicular cancer. However, rRPLND offers low morbidity and improved perioperative outcomes while maintaining the oncologic efficacy of the open approach. In a recent propensity score-matched analysis of patients who underwent primary RPLND, after adjusting for baseline characteristics at a median follow-up period of 23.5 months, the relapse rate was similar between groups: 3.8% in the rRPLND group and 7.8% in the o-RPLND group, with no statistically significant difference (hazard ratio 0.65, 95% CI 0.07–6.31, *p* = 0.70) [30]. Notably, there were no relapses within the surgical field in either group. Moreover, in terms of perioperative outcomes, rRPLND was associated with a significantly shorter hospital stay (1 day vs. 5 days, *p* < 0.0001) and less blood loss (200 mL vs. 300 mL, *p* = 0.032), though it required a longer operative time (8.8 h vs. 4.3 h, *p* < 0.0001) [30].

According to EAU guidelines, early-stage NSGCT patients should undergo a nerve-sparing procedure to maintain antegrade ejaculation, thus reducing the long-term sexual dysfunction morbidity. In a robotic cohort, Pearce et al. reported a 100% preservation of antegrade ejaculation [31]. Similar excellent functional outcomes have been observed in smaller rRPLND series [32]. Interestingly, in a comparative study with laparoscopic RPLND, 11% of patients who underwent the laparoscopic procedure experienced ejaculatory dysfunction, whereas no events were reported in the robotic cohort [33].

Controversies arise due to the procedure’s extent and the complications associated with REj, prompting various modifications. These modifications have led to the development of unilateral RPLND templates. Regarding the post-chemotherapy setting, several studies reported no significative differences between a bilateral vs. unilateral approach, with recurrences ranging from 0 to 10% for both approaches [34,35,36]. However, the limited research and varying results in this area indicate a need for further studies to explore possible predictors of these outcomes.

Despite its novelty, our study is not exempt from limitations. Firstly, considering the intrinsic retrospective nature of data, some reliable and unreliable information is missing. Indeed, the psychological surgical aspect plays a crucial role in the development of sexual dysfunctions and could explain the high rate of ED that we found at 12 months after the procedure [37]. Second, Unfortunately, a direct comparison between the current results and data from our previous open series was not feasible. This is due to the lack of historically prospectively collected specific data for comparison purposes. Third, due to the numerosity of the included population, we could not run further statistical analysis to disclose predictors of sexual dysfunction in men undergoing unilateral PC-rRPLND. Consequently, we encourage further research to fully elucidate this aspect to better assist and meet patients’ expectations.

## 5. Conclusions

About a third of patients undergoing PC-rRPLND for stage II NSGCTs report postoperative onset of sexual dysfunction. Among them, a non-negligible number of patients experience REj or ED. Considering the increasing survival rates and life expectancy of these patients, we encourage further research in the surgical field to prevent andrological complications and to better counsel patients undergoing this life-changing procedure.

## Figures and Tables

**Table 1 cancers-16-02231-t001:** Perioperative characteristics of the population.

Variable	
**Age, year**	32 (26–39)
**BMI, kg/m^2^, median (IQR)**	25 (23–28)
**ASA score, n (%)**	
I–II	20 (91.7)
III–IV	2 (8.3)
**Primary tumor laterality, n (%)**	
Right	10 (45.4)
Left	12 (54.6)
**Pre-chemotherapy clinical stage, n (%)**	
IIA	1 (4.5)
IIB	15 (68.2)
IIC	6 (27.3)
**Rete testis invasion, n (%)**	11 (50)
**LVI, n (%)**	15 (68.2)
**>70% proliferation rate, n (%)**	11 (50.0)
**>50% embryonal carcinoma, n (%)**	12 (54.5)
**Teratoma, n (%)**	8 (36.4)
**Pre-operative IIEF-5, median (IQR)**	24 (23–25)
**Pre-operative EHS, median (IQR)**	4 (4–4)
**Pre-chemotherapy node size, mm**	34 (21–55)
**Post-chemotherapy node size, mm**	23 (19–35)
**Follow-up, months, median (IQR)**	36 (20–42)

Abbreviations: BMI: body mass index, ASA: American Society of Anesthesiologists; LVI: lymphovascular invasion; IQR: interquartile range; IIEF-5: International Index of Erectile Function-5; EHS: Erection Hardness Score.

**Table 2 cancers-16-02231-t002:** Sexual outcomes.

	n (%)
**Andrological disorder**	7 (31.8%)
REj	3 (13.6%)
ED	3 (13.6%)
Hypospermia	1 (4.5%)
**Achievement of successful conception**	2 (9.1%)

REj: retrograde ejaculation; ED: erectile dysfunction.

## Data Availability

The data can be shared up on request.

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
