# Peer review of "Unilateral Post-Chemotherapy Robot-Assisted Retroperitoneal Lymph Node Dissection for Stage II Non-Seminomatous Germ Cell Tumors: Sexual and Reproductive Outcomes"

_cancers, 2024, doi:10.3390/cancers16122231_

Round 1
Reviewer 1 Report
Comments and Suggestions for Authors
Thank you for inviting me to review the article titled “Unilateral post-chemotherapy robot-assisted retroperitoneal 2 lymph node dissection for Stage II non-seminomatous germ cell tumour: Sexual and reproductive outcomes”
In this Manuscript, the authors report sexual and reproductive outcomes following post-chemotherapy robot-assisted retroperitoneal unilateral lymph node dissection (PC-rRPLND) for non-seminomatous germ cell tumors (NSGCT) at a single institution.
Criticisms/comments:
· Any particular reason why the time frame chose of 2018-2021 was chosen that way?
· Authors should explain basic component of IIEF -EF scoring and why it was chosen.
· Only patients with pre-operative IIEF-Erectile Function (IIEF-EF) scores ≥22 and 82 EHS scores ≥3 were included for further analysis – why is this, authors should explain.
· It would be important to know the total number of patients who had RPLND by any method for stage II GCT during the timeframe chosen to give an idea of how many (percentage of) patients fit the study criteria from the total population.
· Usually first paragraph in the discussion summarizes the study results
Comments on the Quality of English Languagenone
Author Response
Dear reviewer, first of all thank you for your observations.
Attached our reply

Reviewer 2 Report
Comments and Suggestions for Authors
This study was reported sexual and reproductive outcomes of unilateral robot-assisted retroperitoneal lymph node dissection for stage II non-seminomatous germ cell tumor after chemotherapy. The evidence is weak, and overall it does not conform to the author guidelines. It needs significant revision.
Major point
1. The author should clearly outline the advantages of unilateral LND compared to bilateral LND.
2. The author should demonstrate more specifically the advantages of Robot compared to Laparo/Open.
3. The author should illustrate the extent of LND and port placement.
4. The author should statistically emphasize sexual and reproductive outcomes. If you cannot achieve, it would be premature to report in the original article.
5. Citations should be listed in accordance with the author guidelines.
Minor point
Abstract ro-bot→robot
Inter-national→International
Line 52: double space
Author Response
dear reviewer, thank you for your valuable suggestions, they certainly improved our manuscript.
Please find attached our reply letter.
warmest regards

Reviewer 3 Report
Comments and Suggestions for Authors
The authors present data on sexual and reproductive function with post-chemotherapy robot-assisted retroperitoneal unilateral lymph node dissection (PC-rRPLND) in stage II nonseminoma germ cell tumors. As mentioned by the authors, there have been few studies presenting such data, and they are of great interest in understanding postoperative sexual and reproductive function in young patients, and are worthy of some reporting. However, the following points need to be corrected.
Major points:
1. The authors mentioned that the median follow-up period was 36 months, however, How many patients were lost to follow up? If the final outcome is sexual function and conception, we need to know the denominator accurately, and I think that should be stated properly. If there was not a single case of interrupted visits, please describe that.
2. If all patients had IIEF-5 and EHS scores recorded preoperatively, then the authors should present those data (including the values for each domain with respect to IIEF-5). Also, if all patients had these questionnaires recorded at 12 months postoperatively, since the number of patients is not that large, why not show the preoperative and postoperative data, even if it is spider plots?
3. The authors mentioned that hypospermia was observed in one patient after PC-rRPLND, what were the baseline findings of semen analysis for this patient? Without a comparison to baseline, I don't think it is possible to say that it was really triggered by the surgery. If any testing did not underwent at baseline, I think that fact should be properly noted in the limitation.
4. If it is valuable that two patients led to pregnancy, why don't you show in a separate Table the clinical characteristics of those two patients (primary tumor laterality, histology, node size before and after chemotherapy, time from surgery to pregnancy confirmation, etc.) so that the readers are informed?
5. In the Materials and Methods section, the description of statistical analysis states that “two-sided” was performed with a p-value <0.05 as significant.
In addition, in the Results section, I do not see any data presented with a two-sided test. If a two-sided test was not performed, the contents of the Materials and Methods section should be properly described.
Author Response
Dear reviewer, thanks for your insight comment. In the attached file our point-by-point reply.
With Regards

Round 2
Reviewer 2 Report
Comments and Suggestions for Authors
Dear authors
I think it was well revised.
Minor
1. In table1, the authors should add IQR in age and BMI as the other factors.
2. In discussion paragraph, spaces after new paragraph should be standardized.
Author Response
Dear reviewer, we are glad that our revisions have been received positively. This feedback is a testament to our team's hard work and dedication.
Minors were made according to your round 2 suggestion.
Warmest Regards
Ty again for you time
Reviewer 3 Report
Comments and Suggestions for Authors
Overall, the manuscript has been revised well and is likely significant and generally supported by the data. I think this manuscript will be acceptable after some corrections have been done.
Minor points:
The abbreviation for retrograde ejaculation is listed as a first appearance, even though it has already appeared (page 6, line 229). Please include the abbreviation in the relevant section.